# Genome-Wide Insights into Intraspecific Taxonomy and Genetic Diversity of Argali (*Ovis ammon*)

Arsen Dotsev [1,*], Olga Koshkina [1], Veronika Kharzinova [1], Tatiana Deniskova [1], Henry Reyer [2], Elisabeth Kunz [3], Gábor Mészáros [4], Alexey Shakhin [1], Sergey Petrov [1], Dmitry Medvedev [5], Alexander Kuksin [6], Ganchimeg Bat-Erdene [5,7], Bariushaa Munkhtsog [8], Vugar Bagirov [1], Klaus Wimmers [2], Johann Sölkner [8], Ivica Medugorac [3] and Natalia Zinovieva [1,*]

[1] L.K. Ernst Federal Research Center for Animal Husbandry, 142132 Podolsk, Russia; olechka1808@list.ru (O.K.); veronika0784@mail.ru (V.K.); horarka@yandex.ru (T.D.); alexshahin@mail.ru (A.S.); citelekle@gmail.com (S.P.); vugarbagirov@mail.ru (V.B.)
[2] Institute of Genome Biology, Research Institute for Farm Animal Biology (FBN), 18196 Dummerstorf, Germany; reyer@fbn-dummerstorf.de (H.R.); wimmers@fbn-dummerstorf.de (K.W.)
[3] Population Genomics Group, Department of Veterinary Sciences, Ludwig-Maximilians-University Munich, 80539 Munich, Germany; elisabeth.kunz@gen.vetmed.uni-muenchen.de (E.K.); ivica.medjugorac@gen.vetmed.uni-muenchen.de (I.M.)
[4] Institute of Livestock Sciences, University of Natural Resources and Life Sciences (BOKU), 1180 Vienna, Austria; gabor.meszaros@boku.ac.at
[5] Department of Game Management and Bioecology, Irkutsk State University of Agriculture, 664038 Molodezhny, Russia; dmimedvedev@yandex.ru (D.M.); b.ganaa1205@gmail.com (G.B.-E.)
[6] Tuva Institute for the Integrated Development of Natural Resources of the Siberian Branch of the Russian Academy of Sciences, 667007 Kyzyl, Russia; kuksintuva@yandex.ru
[7] Khovd Branch School of National University of Mongolia, Jargalant 67301, Mongolia
[8] Irbis Mongolia Center, Ulaanbaatar 13380, Mongolia; mtsogb@gmail.com (B.M.); johann.soelkner@boku.ac.at (J.S.)
* Correspondence: asnd@mail.ru (A.D.); n_zinovieva@mail.ru (N.Z.); Tel.: +7-4967651104 (A.D.); +7-4967651404 (N.Z.)

**Abstract:** Argali (*Ovis ammon*), the largest species among all wild sheep, is native to mountainous regions of Central and East Asia, spreading mainly throughout such countries as Tajikistan, Kyrgyzstan, Kazakhstan, Afghanistan, Mongolia, Russia and China. Intraspecific taxonomy of argali remains unclear, and currently, most researchers recognize up to nine subspecies. The aim of our work was to investigate the phylogenetic relationship between populations of *O. ammon* based on genome-wide SNP analysis. Five subspecies, Altai (*O. a. ammon*) (*n* = 6), Gobi (*O. a. darwini*) (*n* = 5), Pamir (*O. a. polii*) (*n* = 12), Tian Shan (*O.a. karelini*) (*n* = 15) and Kyzylkum (*O. a. severtzovi*) (*n* = 4), were genotyped using Illumina OvineHD BeadChip. In addition, complete mitogenome sequences from 30 of those samples were obtained. After quality control procedures, 65,158 SNPs were selected for the subsequent analyses. Neighbor-Net dendrogram and principal component analysis (PCA) revealed that the five subspecies could be grouped into four clusters. It was shown that a population from Altai formed a cluster with Gobi subspecies. The highest pairwise $F_{ST}$ genetic distance was between *O. a. ammon* and *O. a. severtzovi* (0.421) and the lowest were between *O. a. polii* and *O.a. karelini* (0.083) and between *O. a. ammon* and *O. a. darwini* (0.040) subspecies. Genetic diversity was higher in Central Asian argali as compared to East Asian populations. *O. a. severtzovi* had an admixed origin and consisted of two genetic components—73.5 ± 0.2% of *O. a. polli* and 26.5 ± 0.2% of urial (*O. vignei*). TreeMix analysis revealed a migration event from urial to *O. a. severtzovi* argali. The analysis of complete mitogenomes supported the results based on whole-genome genotyping. Considering that all the mtDNA sequences of *O. a severtzovi* belonged to *O. ammon* and not to *O. vignei*, we concluded that gene flow in this group was associated with urial males. As this is only the first work on intraspecific taxonomy and genetic diversity of argali based on genome-wide SNP genotyping and the analysis of complete mitogenomes, we suggest that more genetic studies are needed to clarify the status of Gobi and Tian Shan argali.

**Keywords:** genetic diversity; population structure; wild sheep; SNP; mitochondrial DNA; phylogenetics

## 1. Introduction

Argali (*Ovis ammon*) is the largest species among all wild sheep that can weigh over 200 kg, have a body length up to two meters and be up to 137 cm in height [1]. The range of *O. ammon* includes mountainous areas in Central and East Asia, and spreads out from the Altai Mountains (Russia, Mongolia) and Niyaz range (Kazakhstan) in the north to the Himalaya mountains (China, Nepal) in the south, and from Nuratau ridge (Uzbekistan) in the west to the Khentii mountains (Mongolia) in the east.

The chromosome diploid number in all the studied populations of argali is 56 and that distinguishes this species from the other representatives of the genus Ovis. Snow sheep (*O. nivicola*) has $2n = 52$ chromosomes; bighorn (*O. canadensis*), thinhorn (*O. dalli*), mouflon (*O. gmelini*), domestic sheep (*O. aries*) have $2n = 54$ chromosomes; and urial (*O. vignei*) has $2n = 58$ chromosomes. According to Sanna et al. [2,3], *O. ammon* diverged from its closest relative species—*O. gmelini* and *O. vignei*—1.11 million years ago (MYA) and from the other *Ovis* group which includes Siberian snow sheep (*O. nivicola*) as well as North American bighorn (*O. canadensis*) and thinhorn (*O. dalli*) 2.66 MYA.

The most fundamental intraspecific classifications of argali were proposed by Lydekker in 1898 [4], Nasonov in 1923 [5], Tsalkin in 1951 [6], Nadler et al. in 1973 [7], Schaller in 1977 [8], Valdez in 1982 [9] and Geist in 1991 [10]. In those works, from 4 to 16 subspecies were recognized. Currently most researchers [11,12] agree with the taxonomy proposed by the IUCN Caprinae Specialist Group in 1997 [13], in which they recognized nine subspecies of argali (*O. ammon*): Altai (*O. a. ammon*), Gobi (*O. a. darwini*), Pamir or Marco Polo sheep (*O. a. polii*), Tian Shan (*O. a. karelini*), Kyzylkum or Severtsov's sheep (*O. a. severtzovi*), Northern Chinese (*O. a. jubata*), Tibetan (*O. a. hodgsonii*), Kazakhstan (*O. a. collium*) and Karatau (*O. a. nigrimontana*).

In our work, we investigated five subspecies of argali: *O. a. ammon* from Russia, *O. a. darwini* from Mongolia, *O. a. polli* from Tajikistan, *O. a. karelini* from Kyrgyzstan and *O. a. severtzovi* from Uzbekistan using genome-wide SNP analysis and complete mitochondrial genomes. Two samples of *O. a. karelini* were collected in the area of Kyrgyz argali (*O. a. humei*) habitat, the subspecies that was described by Lydekkker in 1913, but is considered invalid by most researchers [1]. Since the habitat of this group of argali is located at the intersection of *O. a. karelini* and *O. a. polii* ranges, it was usually attributed to either the former [9] or the latter [11]. *O. a. ammon* inhabits the Altai Mountains in Russia and Mongolia. The census size of this subspecies, estimated by WWF in 2019, was 4851 individuals in total (1431 in Russia and 3420 in Mongolia). *O. a. darwini* is distributed patchily in mountains, hills and canyons of the Gobi Desert and Gobi steppe in Mongolia [14]. The surveys indicated that approximately 10,000–12,000 individuals belong to this subspecies. *O. a. polii* inhabits the Pamir Alai Mountain System in Tajikistan and adjacent regions of Kyrgyzstan, Afghanistan and China. The estimated population size in the Eastern Pamirs in the survey conducted in December of 2009 was 23,711 argali in 510 herds [15]. A total of 4013 animals attributed to this subspecies were revealed in 2017 in Kyrgyzstan. Tian Shan argali (*O. a. karelini*) is distributed across the Tian Shan Mountains in Kazakhstan, Kyrgyzstan and China [12]. The estimated census size of this population in 2017 was around 12,500 individuals in Kyrgyzstan and around 2500 in Kazakhstan. *O. a. severtzovi* is the smallest in size among all the subspecies of argali. Its current habitat is limited to Nuratau ridge and Central Kyzylkum Mountains in Uzbekistan. According to the IUCN Red List of Threatened Species, less than 2000 animals of this subspecies persist in Uzbekistan. A small number of *O. a. severtzovi* were also recorded in the territories of Kyrgyzstan and Tajikistan bordering Uzbekistan [16].

In recent years, more and more refinements in the taxonomy of animals have been made by molecular genetic studies. Different mitochondrial and nuclear DNA markers

were used to clarify phylogenetic relationships in ungulate species [17–23]. Until now, molecular genetic studies of argali have been carried out using mitochondrial DNA [23,24] and microsatellites [25–27].

Currently, the use of SNP chips developed for domestic species is one of the promising methods for studying their wild relatives. For example, the SNP chip developed for domestic sheep has been successfully used in the research of wild *Ovis* species: *O. canadensis* [28], *O. dalli* [29,30], *O. aries musimon* [31] and *O. nivicola* [20]. The analysis of mitochondrial DNA has been successfully implemented in phylogenetic studies for years [2,17–19,21,32]. Unlike nuclear, mitochondrial DNA is characterized by maternal inheritance, high mutation rate and lack of recombination. Such differences in these genetic markers can lead to dissimilar conclusions on the relationships of both individuals and populations. The combined use of nuclear and mitochondrial DNA gives a more complete picture of evolution.

The aim of our work was to investigate the phylogenetic relationship between populations of *O. ammon* based on genome-wide SNP analysis and complete mitochondrial genomes.

## 2. Materials and Methods

### 2.1. Sample Collection and Ethic Statement

In our study, we examined 42 specimens of argali belonging to five subspecies: *O. a. polii* (*n* = 12), *O. a. karelini* (*n* = 15), *O. a. severtzovi* (*n* = 4), *O. a. ammon* (*n* = 6) and *O. a. darwini* (*n* = 5). Ten specimens of urials (*O. vignei*) from Iran (*O. v. arkal*, *n* = 4) and Pakistan (*O. v. blanfordi*, *n* = 3), (*O. v. punjabienis*, *n* = 3) were included in the dataset as an outgroup. The sampling sites of the studied specimens are presented in Figure 1. Muscle tissue samples of *O. ammon* and *O. vignei* were obtained during scientific expeditions and from trophy hunters, who were officially licensed to hunt argali and urials during hunting season. The protocol for the study, No. 2 (28 April 2022), was approved by the Commission on the Ethics of Animal Experiments of the L.K. Ernst Federal Science Center for Animal Husbandry.

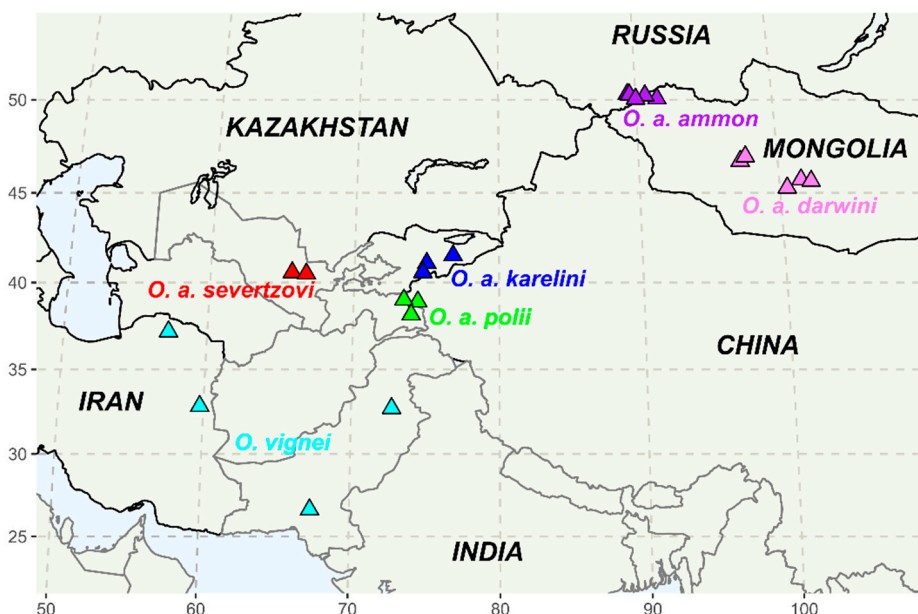

**Figure 1.** Sampling sites of the studied specimens of argali (*O. ammon*) and urial (*O. vignei*).

### 2.2. Data Quality Control and Processing of Whole-Genome SNP Data

DNA extraction was carried out using Nexttec columns (Nexttec Biotechnology GmbH, Leverkusen, Germany). The genotyping was performed using Illumina Ovine HD Beadchip [33]. SNP quality control filtering was performed in PLINK 1.9 [34] software. Only autosomal SNPs with known positions were selected. SNPs that were genotyped in less than 90% of individuals (−−geno 0.1), with a minor allele frequency (MAF) less than 5%

(−−maf 0.05) and in linkage disequilibrium (−−indep-pairwise 50 5 0.5) were pruned. PLINK 1.9 was also used to perform principal component analysis (PCA) (−−pca 4) and calculate pairwise identity-by-state (IBS) distances (−−distance 1-ibs). Pairwise $F_{ST}$ [35] and Reynolds genetic distances [36] were calculated in the R packages "StAMPP" [37] and "adegenet" [38], respectively. Neighbor-Net dendrograms were constructed in SplitsTree 4.14.6 program [39]. Visualization of PCA was generated with the R package "ggplot2" (ver. 3.3.2) [40]. The maximum likelihood phylogenetic tree was constructed using Treemix 1.13 software [41]. We used a block size of 500 SNPs and ran ten iterations of 0 to 9 migration events. The optimum number of migration events was found using the R package "OptM" [42]. Cluster analysis was performed in the program Admixture 1.3 [43] with visualization in the R package "pophelper" [44].

Genetic diversity characteristics including allelic richness, observed and unbiased expected heterozygosity were calculated in the R package "diveRsity" [45]. Multilocus heterozygosity (MLH) was calculated in the R package "inbreedR" [46].

### 2.3. Processing of Complete Mitogenomes

Using NGS technology, we obtained 30 complete mitochondrial genomes of *O. ammon—O. a. polii* (*n* = 10), *O. a. karelini* (*n* = 7), *O. a. severtzovi* (*n* = 4), *O. a. ammon* (*n* = 4), *O. a. darwini* (*n* = 5)—and five mitogenomes of *O. vignei—O. v. arkal* (*n* = 1), *O. v. blanfordi* (*n* = 2), *O. v. punjabiensis* (*n* = 2). To perform phylogenetic analysis, we used sequences of concatenated 2 rRNA and 13 protein coding genes that were extracted from complete mitochondrial genomes and consisted of 13,879 bp. For the alignment of the sequences, the program Muscle 3.8.31 was used [47].

The best evolution model was determined in the software Partitionfinder 2 [48]. The Bayesian phylogenetic tree was constructed in the program MrBayes 3.2.6 [49]. The analysis was carried out running 10,000,000 Markov chain Monte Carlo (MCMC) generations, sampling every 500 generations with a burn-in of 25%. The visualization was performed in the FigTree 1.4.2 program [50].

### 2.4. The map of Sampling Sites

The map of sampling sites was created with R packages "maps" [51] and "ggplot2".

## 3. Results

### 3.1. Whole-Genome SNP Analysis

After quality control filtering, 42 argali samples and 65,158 SNPs were selected for the subsequent analyses. The principal component analysis (Figure 2A) showed that the five studied populations formed four clusters: (1). *O. a. ammon* and *O. a. darwini*; (2). *O. a. karelini*; (3). *O. a. polii*; (4). *O. a. severtzovi* subspecies. The first component divided East Asian populations—*O. a. ammon* and *O. a. darwini* (PC1 < 0)—from Central Asian—*O. a. polii*, *O. a. karelini* and *O. a. severtzovi* (PC1 > 0). The second component separated *O. a. polii* and *O. a. karelini* from *O. a. severtzovi*. The separation of *O. a. polii* and *O. a. karelini* subspecies was observed at PC3 (Figure S1).

Neighbor-Net dendrogram based on IBS distances (Figure 2B) demonstrated that O. *a. polli*, *O. a. karelini*, *O. a. severtzovi* as well as *O. vignei*, taken as an outgroup, each formed their own separate clusters. In contrast, *O. a. ammon* and *O. a. darwini* samples were placed in the same cluster.

Calculation of cross-validation (CV) error (Figure S2) for the admixture analysis run for values of K from 1 to 10 indicated that the optimal number of populations in our dataset was determined as three (K = 3). At K = 3 (Figure 2C), *O. ammon* was divided into two groups—East Asian and Central Asian. East Asian argali—*O. a. ammon* and *O. a. darwini*—had identical genetic components. In Central Asian populations, only *O. a. polli* was not admixed. *O. a. karelini* had 97.5 ± 0.2% of genetic components of *O. a. polli* and 2.4 ± 0.2% of East Asian argali. *O. a. severtzovi* also consisted of two components—73.5 ± 0.2% of *O. a. polli* and 26.5 ± 0.2% of *O. vignei*. The separation of *O. a.*

*polli* and *O. a. karelini* was observed at K = 5 (Figure S3). Starting from K = 4, *O. a. severtzovi* was presented as a non-admixed population.

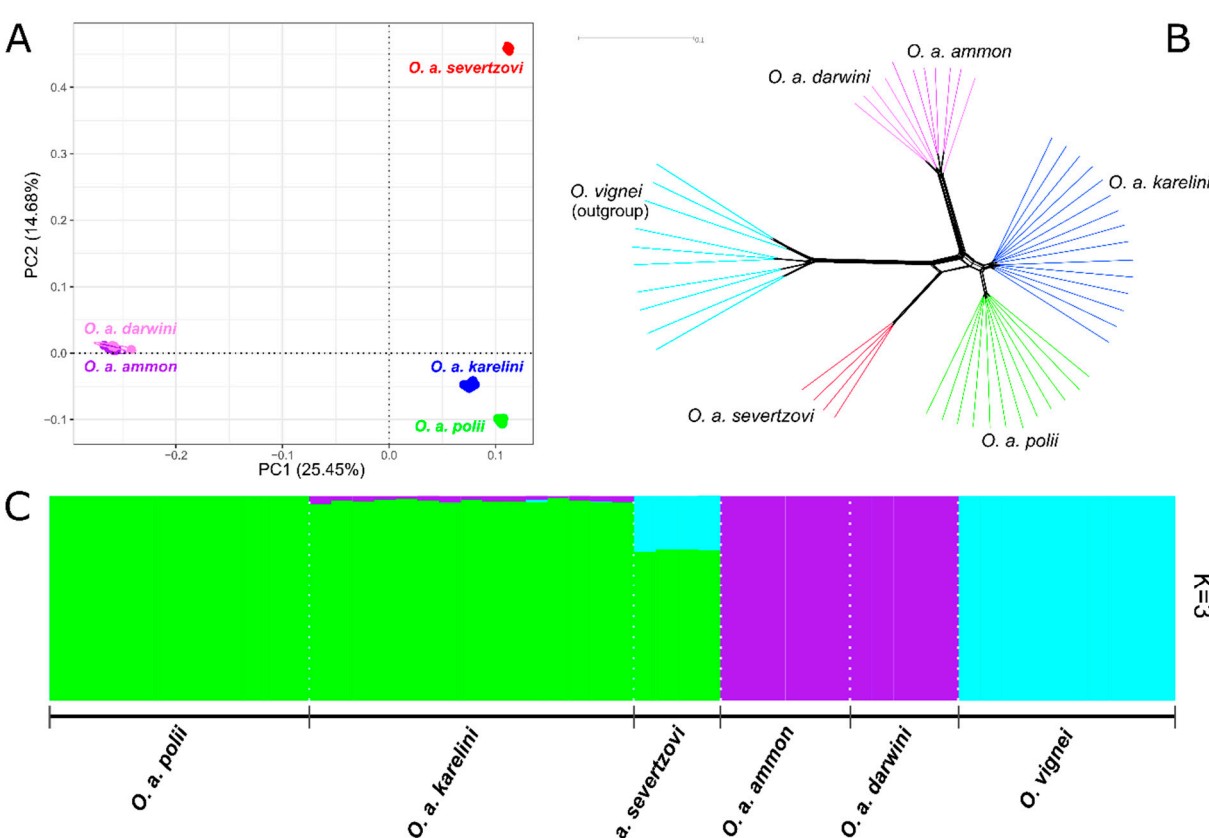

**Figure 2.** (**A**)—Principal component analysis of the studied populations of *O. ammon*; (**B**)—Neighbor-Net dendrogrm of the studied populations of *O. ammon* with *O. vignei* taken as an outgroup; (**C**)—Admixture analysis of the studied populations of *O. ammon* and *O. vignei*.

Pairwise $F_{ST}$ and Reynolds genetic distances were calculated to estimate the differentiation between the studied populations of argali (Table 1). The lowest values for both distances were observed between *O. a. ammon* and *O. a. darwini*, and *O. a. polli* and *O. a. karelini*. The highest genetic divergence was found between East Asian populations of argali and *O. a. severtzovi*. Both $F_{ST}$ and Reynolds values revealed that *O. a. ammon* and *O. a. darwini* were genetically closer to the separate species *O. vignei* than to *O. a. severtzovi*.

**Table 1.** Pairwise $F_{ST}$ (below diagonal) and Reynolds (above diagonal) genetic distances between the groups of argali (*O. ammon*) and urial (*O. vignei*).

|  | *O. a. polii* | *O. a. karelini* | *O. a. severtzovi* | *O. a. ammon* | *O. a. darwini* | *O. vignei* |
|---|---|---|---|---|---|---|
| *O. a. polii* | 0.000 | 0.346 | 0.581 | 0.592 | 0.592 | 0.609 |
| *O. a. karelini* | 0.083 | 0.000 | 0.557 | 0.560 | 0.561 | 0.599 |
| *O. a. severtzovi* | 0.266 | 0.237 | 0.000 | 0.696 | 0.695 | 0.639 |
| *O. a. ammon* | 0.290 | 0.251 | 0.421 | 0.000 | 0.377 | 0.679 |
| *O. a. darwini* | 0.284 | 0.244 | 0.414 | 0.040 | 0.000 | 0.677 |
| *O. vignei* | 0.339 | 0.333 | 0.325 | 0.397 | 0.386 | 0.000 |

The migration events for the populations of *O. ammon* were assessed by constructing a maximum likelihood tree as implemented in the Treemix analysis (Figure 3). To infer the optimum number of migration events, the Treemix results were evaluated using the Evanno method which revealed a maximum value for delta m of 14.8122 at m = 2 edges

(Figure S6). To select the best-fit tree, the residual fit plot was created (Figure S7). This allowed us to check if all the parts of the tree had been well modeled. The first migration event we observed was from a ghost population or the unknown ancestor of all *O. ammon* subspecies to *O. a. severtzovi* ($p = 2.22 \times 10^{-308}$). The second migration event ($p = 0.049$) with less weight revealed a gene flow from *O. vignei* to the common ancestor of *O. a. polii* and *O. a. severtzovi*.

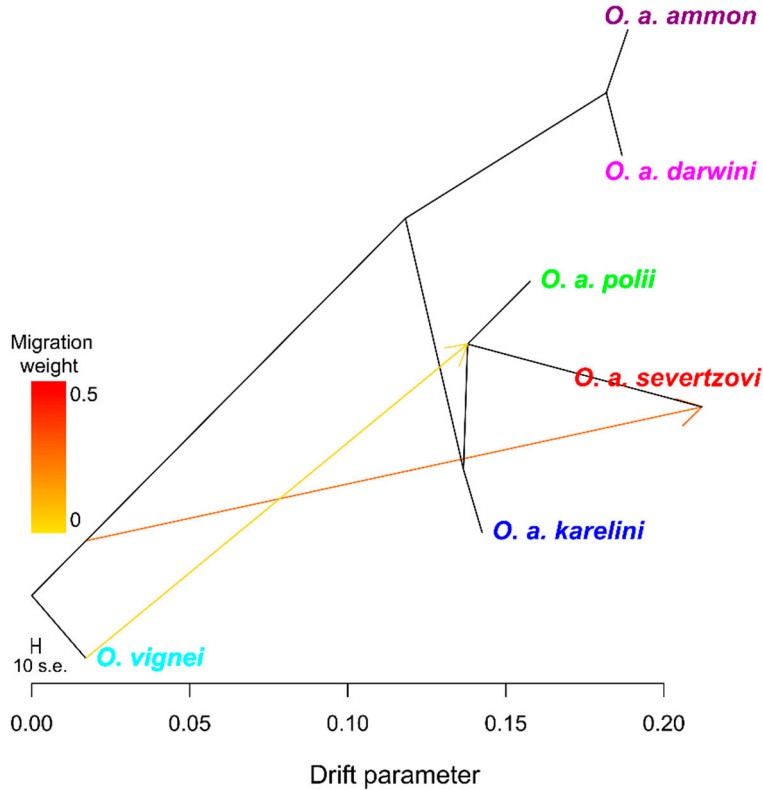

**Figure 3.** Maximum likelihood phylogenetic tree with two migration events, which are indicated by the arrows. Migration arrows are colored according to their weight.

The highest values of genetic diversity were observed in Central Asian argali populations, composed of *O. a. karelini* and *O. a. polii*. The unbiased expected heterozygosity in the above-mentioned subspecies was $0.295 \pm 0.001$ and $0.286 \pm 0.001$, and allelic richness was $1.714 \pm 0.001$ and $1.685 \pm 0.001$, respectively (Table 2). In the East Asian populations, all the genetic diversity parameters were significantly lower—unbiased expected heterozygosity did not exceed $0.208 \pm 0.001$ and allelic richness ranged from $1.484 \pm 0.002$ to $1.499 \pm 0.002$. The inbreeding coefficient (Fis) was positive in all the studied populations. The deficit of heterozygotes was lower in Central Asian argali—from 0.011 in *O. a. severtzovi* to 0.024 in *O. a. polii*—than in East Asian populations—from 0.092 in *O. a. darwini* to 0.128 in *O. a. ammon*.

**Table 2.** Genetic diversity characteristics of the studied populations of *O. ammon* and *O. vignei*.

| Population | $n$ | Ho ± se | uHe ± se | Fis [CI 95%] | $A_R$ ± se | MLH Mdn [Range] |
|---|---|---|---|---|---|---|
| *O. a. polii* | 12 | 0.277 ± 0.001 | 0.286 ± 0.001 | 0.024 [0.022; 0.026] | 1.685 ± 0.001 | 0.279 [0.212; 0.290] |
| *O. a. karelini* | 15 | 0.289 ± 0.001 | 0.295 ± 0.001 | 0.019 [0.017; 0.021] | 1.714 ± 0.001 | 0.288 [0.259; 0.299] |
| *O. a. severtzovi* | 4 | 0.242 ± 0.001 | 0.245 ± 0.001 | 0.011 [0.007; 0.015] | 1.570 ± 0.002 | 0.240 [0.237; 0.248] |
| *O. a. ammon* | 6 | 0.173 ± 0.001 | 0.203 ± 0.001 | 0.128 [0.124; 0.132] | 1.484 ± 0.002 | 0.179 [0.110; 0.191] |
| *O. a. darwini* | 5 | 0.185 ± 0.001 | 0.208 ± 0.001 | 0.092 [0.088; 0.096] | 1.499 ± 0.002 | 0.180 [0.171; 0.208] |
| *O. vignei* | 10 | 0.312 ± 0.001 | 0.356 ± 0.001 | 0.111 [0.108; 0.114] | 1.813 ± 0.001 | 0.313 [0.268; 0.329] |

Notes: *n*—number of samples, Ho—observed heterozygosity, uHe—unbiased expected heterozygosity, se—standard error, Fis—inbreeding coefficient, CI 95%—95% confidence interval, Ar—allelic richness, MLH—multilocus heterozygosity, Mdn—median.

Multilocus heterozygosity (MLH) values, which reflect individual genome-wide heterozygosity, ranged from 0.110 in one specimen of *O. a. ammon*, which could be considered an outlier, to 0.299 in a specimen of *O. a. karelini*. Graphical visualization of MLH is presented in Figure 4.

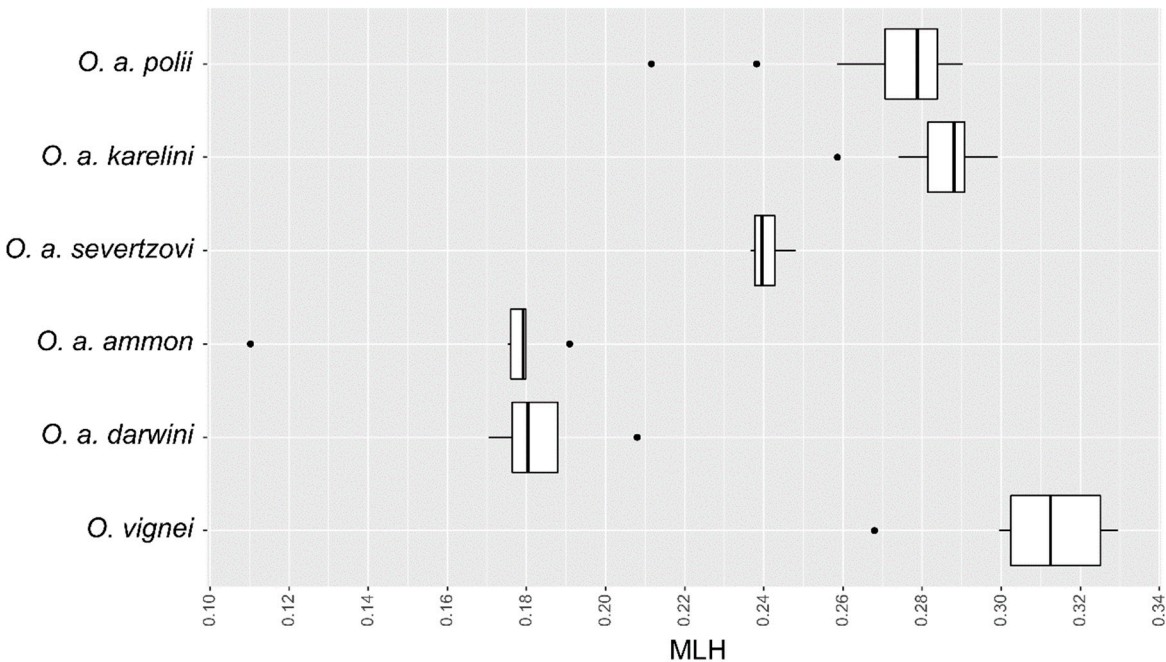

**Figure 4.** Boxplot indicating multilocus heterozygosity (MLH) in the studied populations of *O. ammon* and *O. vignei*.

*O. vignei* samples were included in this analysis to identify the variance in heterozygosity among individuals of the species which was taken as an outgroup. The comparison of *O. ammon* and *O. vignei* is not being discussed as these results might have been influenced by ascertainment bias.

### 3.2. Complete Mitochondrial Genomes Analysis

The analysis of complete mitochondrial genomes was conducted by constructing the Bayesian phylogenetic tree (Figure 5) based on concatenated sequences of 2 rRNAs and 13 protein-coding genes. All the clades in the tree were strongly supported by high posterior probability values, which varied from 0.97 to 1.

East Asian argali belonged to two different groups. The clade that had more a distant common ancestor with Central Asian populations contained three sequences of *O. a. ammon* and two sequences of *O. a. darwini*. The other clade included one and three sequences of those subspecies, respectively. The Central Asian populations clustered separately from each other. All the *O. a. severtzovi* specimens had almost identical haplotypes and they were genetically closer to *O. a. polli*. Two different subclades could be distinguished in the *O. a. karelini* cluster. We did not observe any *O. vignei* haplotypes among the studied sequences of *O. ammon*.

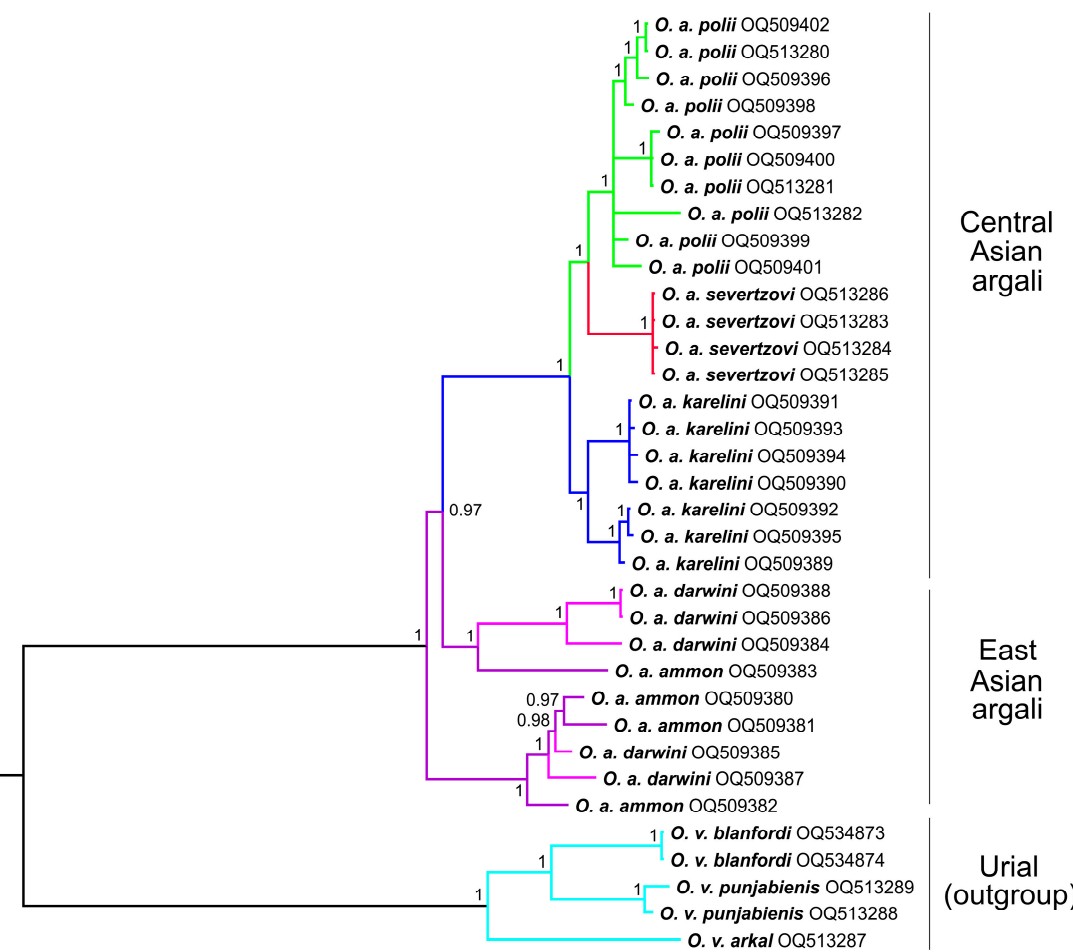

**Figure 5.** Rooted Bayesian phylogenetic tree of the studied populations of *O. ammon* and *O. vignei* (outgroup) based on complete mitochondrial genome sequences. Posterior probability values are indicated on the nodes.

## 4. Discussion

Here, we present the first work on genetic diversity and intraspecific relationships between populations of *O. ammon* based on genome-wide SNP genotyping and the analysis of complete mitochondrial genomes. The genome-wide genotyping of 42 specimens belonging to 5 subspecies allowed the selection of more than 65 thousand SNPs. To corroborate the results inferred by the analyses carried out on the nuclear DNA, 30 mitogenome sequences from the studied specimens were included in our research.

The results of our study pointed out high genetic differentiation between East Asian and Central Asian groups of argali. *O. a. severtzovi* had high $F_{ST}$ values with all the other studied populations—from 0.237 (*O. a. karelini*) to 0.421 (*O. a. ammon*). These results agree with the taxonomy based on morphology [9,10,52,53]. Such an increased genetic distance could also be the result of admixture with foreign subspecies.

Genetic diversity values, such as allelic richness, observed and expected heterozygosity, did not differ much in the neighboring groups. In East Asian argali, these parameters were much lower than in Central Asian populations. Similar results were reported in the study of Tserenbataa et al. [24] where the comparison of nucleotide diversity in populations of argali from Mongolia, Kazakhstan and Kyrgyzstan was made. We suggest that harsh climatic conditions and partial glaciation of the Altai Mountains during the last glacial maximum could lead to a decrease in genetic diversity in East Asian populations of argali.

### 4.1. The Phylogenetic Relationship between O. a. ammon and O. a. darwini

The phylogenetic relationship between *O. a. ammon* and *O. a. darwini* was previously studied using phenotypic traits, mitochondrial DNA and microsatellites. Based on craniological features, body proportions and color variation, Valdez [9], Sopin [52], Geist [10] and Kapitanova et al. [53] confirmed the subspecies rank of *O. a. ammon* and *O. a. darwini*. In the study of the mitochondrial control region and 14 microsatellite loci of argali from the Altai Mountains, Hangay Mountains and eastern Gobi Desert, Feng et al. [25] indicated genetic differentiation among these groups. Using mitochondrial control region hypervariable segment sequence (598 bp) and three microsatellite markers Delgerzul et al. [27] stated that the genetic difference between Altai and Gobi populations was at the subspecies level. The results of our work agree with the study of Tserenbataa et al. [24], where using the ND5 gene of mitochondrial DNA, the authors revealed low genetic differentiation and a high level of gene flow among populations and concluded that *O. a. ammon* and *O. a. darwini* should be considered a single subspecies. In our work, the analysis of more than 65 thousand SNPs located throughout the genome, did not reveal the separation of these groups from each other. All the samples were in the same cluster both in the PCA and on the Neighbor-Net dendrogram. No differences between the individuals were observed in the admixture analysis. Pairwise $F_{ST}$ genetic distances revealed low differentiation between *O. a. ammon* and *O. a. darwini*—0.04. However, the value of Reynolds genetic distances was even higher (0.377), than between *O. a. polii* and *O. a. karelini* argali (0.346). The analysis of complete mitochondrial genomes revealed that *O. a. ammon* and *O. a. darwini* formed two clades. Three samples of *O. a. ammon* belonged to the clade that was genetically more distant from Central Asian argali and one sample to the clade, which had a more recent common ancestor with this group. *O. a. darwini* samples also belonged to these two clades—two and three samples, respectively.

Based on our results, we suggest that additional research with more specimens representing all habitats in Altai and Gobi Mountain systems is needed to determine the taxonomic status of these populations.

### 4.2. The phylogenetic Relationship between O. a. polii, O. a. karelini, and O. a. humei

*O. a. polii* and *O. a. karelini* are usually regarded as different groups even though their ranges overlap. Morphological features characteristic of each of these populations are considered by most researchers to be sufficient to distinguish them into separate subspecies [5,6,9,10,53]. In contrast in the work of Sopin [52], based on the analysis of cranial proportions, the two groups were combined into one subspecies—*O. a. polii*. The analysis of mitochondrial control region (D-loop) [54] demonstrated that specimens from Tajikistan and Kyrgyzstan had unique haplotypes but were all placed in the neighboring clades. In our study of complete mitochondrial genomes, we obtained similar results—ten samples of *O. a. polii* and seven samples of *O. a. karelini* formed separate clusters. The first comparison of these populations on a nuclear genome-wide level, performed in the present study, revealed moderate genetic differentiation ($F_{ST}$ value—0.084). All the studied samples clustered separately within their respective groups in principal component analysis and on the Neighbor-Net dendrogram. In the admixture analysis, *O. a. polii* and *O. a. karelini* are divided from each other at K = 5, a result that supports their classification as distinct subspecies.

Among the *O. a. karelini* samples, two were obtained from habitats of Kyrgyz argali (*O. a. humei*) which most researchers do not recognize as a valid subspecies [1]. In our study, no genetic distinctiveness was found in the *O. a. humei* samples that therefore were regarded as *O. a. karelini*. Based on such results, we suggest that *O. a. humei*, which morphologically resembles both *O. a. polii* and *O. a. karelini*, should not be considered a separate subspecies but should be attributed to *O. a. karelini*.

*4.3. The Evolution and Phylogeny of O. a. severtzovi*

Kyzylkum argali was first described as a separate species—*Ovis severtzovi* in 1914 by Nasonov [55], who later regarded it as a subspecies of argali [5]. During the 20th century, many attempts to classify this group of wild sheep were made. Due to its morphological similarity to both *O. ammon* and *O. vignei*, some researchers attributed it to argali [8], and others to urials [6,9]. Geptner et al. [56], followed by Groves and Grubb [57] suggested the possible hybrid origin of Kyzylkum argali.

In 1998, Bunch et al. [58] performed the cytogenetic investigation and determined that Severtzov's sheep had a diploid chromosome number of $2n = 56$. This makes it consistent with the other subspecies of *O. ammon*, but not with *O. vignei*, whose diploid chromosome number is 58. Wu et al. [23] in 2003 analyzed the sequences of the mitochondrial DNA control region (1206 bp) and concluded that it was reasonable to classify Severtzov's sheep as a subspecies of argali. In 2010, Rezaei et al. [17] conducted a study on the taxonomy of wild species of the genus *Ovis* based on combined mitochondrial DNA—cytochrome b gene (1140 bp) and four nuclear loci—Interleukin 4 (IL4), Kappa casein (CSN3), Keratin associated protein 1.3 (KAP13) and Interleukin 16 (IL16). According to that work, as well as in the above-mentioned molecular genetic studies, Severtzov's sheep was clearly recognized as *O. ammon*.

In our study, the whole genome SNP genotyping of four specimens of Severtzov's sheep allowed us to classify this population as a subspecies of argali that had an admixture with *O. vignei*. We identified that its genome contained about a quarter of urials' genetic components. The hybrid origin was also confirmed by the TreeMix analysis, which revealed a migration event from a ghost population related to *O. vignei* to *O. a. severtzovi*. It should be noted that *O. a. severtzovi* belongs to Central Asian argali, as its pairwise genetic distances, both $F_{ST}$ and Reynolds, with East Asian argali were even higher than with *O. vignei*. The Bayesian phylogenetic reconstruction based on complete mitochondrial genomes demonstrated that all the studied sequences of Severtzov's sheep belonged to *O. ammon*. The comparison of nuclear and mitochondrial DNA-based results allowed us to suggest that gene flow from *O. vignei* to *O. a. severtzovi* was associated with migrations of urial males.

## 5. Conclusions

In our study, we investigated the genetic structure and phylogeny of five groups of *O. ammon*, based on whole-genome SNP analysis and complete mitochondrial genomes. We observed high genetic differentiation of *O. a. severtzovi* from all the other populations and low differentiation between *O. a. ammon* and *O. a. darwini*. Despite the fact that *O. a. polii* and *O. a. karelini* clustered separately, genetic differentiation between them was moderate. *O. a. severtzovi* was found to have an admixed origin and consisted of two genetic components—from *O. a. polli* and *O. vignei*. Mitochondrial DNA analysis revealed that gene flow was associated with urial males. We suggest that more genetic studies are needed to clarify the status of Gobi and Tian Shan argali.

**Supplementary Materials:** The following supporting information can be downloaded at: https://www.mdpi.com/article/10.3390/d15050627/s1, Figure S1: Principle component analysis (PC1 and PC3) of the studied populations of *O. ammon*; Figure S2. Cross validation (CV) error for the Admixture analysis for values of K from 1 to 10; Figure S3. Admixture analysis of the studied populations of *O. ammon* and *O. vignei* for K from 2 to 5; Figure S4. Neighbor-Net dendrogram of the studied populations of *O. ammon* based on $F_{ST}$ pairwise genetic distances; Figure S5. Neighbor-Net dendrogram of the studied populations of *O. ammon* based on Reynolds pairwise genetic distances; Figure S6. The optimal number of migration edges for the Treemix analysis evaluated using the Evanno method; Figure S7. The residual fit plot corresponding to Treemix analysis with two migration events.

**Author Contributions:** Conceptualization, A.D. and N.Z.; methodology, A.D., O.K., T.D., H.R., G.M., E.K., K.W., J.S. and I.M.; software, A.D. and A.S.; validation, A.D. and S.P.; formal analysis, A.D. and A.S.; investigation, A.D.; resources, D.M., A.K., G.B.-E. and B.M.; data curation, A.D. and N.Z.; writing—original draft preparation, A.D.; writing—review and editing, N.Z. and V.K.; visualization, A.D.; supervision, N.Z.; project administration, N.Z. and V.B.; funding acquisition, A.D. All authors have read and agreed to the published version of the manuscript.

**Funding:** This study was supported by the RSF within project No. 21-46-00001 (SNP genotyping) and the RMSHE (DNA collection and mitochondrial DNA sequencing) within project No. FGGN-2022-0002.

**Institutional Review Board Statement:** The study was approved by the Commission on the Ethics of Animal Experiments of the L.K. Ernst Federal Science Center for Animal Husbandry (the protocol No. 2 (28 April 2022)).

**Informed Consent Statement:** Not applicable.

**Data Availability Statement:** SNP data in PLINK format was deposited to Figshare: https://doi.org/10.6084/m9.figshare.22722616. The complete mitochondrial genomes of argali and urials were deposited to NCBI Genbank (https://www.ncbi.nlm.nih.gov/nuccore, accessed on 28 March 2023) under accession numbers OQ509380—OQ509402, OQ513280—OQ513289 and OQ534873—OQ534874.

**Acknowledgments:** We are grateful to Bendersky E.V. and the Mountain Hunters Club (www.kgo-club.ru, accessed on 28 March 2023) for providing the samples of *Ovis ammon* and *Ovis vignei.* We thank our reviewers for the valuable comments and suggestions that greatly improved the manuscript. The equipment of the Center for Biological Resources and Bioengineering of Agricultural Animals (L.K. Ernst Federal Science Center for Animal Husbandry) was used in the research.

**Conflicts of Interest:** The authors declare no conflict of interest.

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
