# Peer review of "Genome-Wide Insights into Intraspecific Taxonomy and Genetic Diversity of Argali (Ovis ammon)"

_diversity, doi:10.3390/d15050627_

Round 1
Reviewer 1 Report
Overall, a well-structured manuscript with interesting results in relation to the intraspecific taxonomy and genetic diversity of argali (Ovis ammon). The “Introduction” provides the reader with sufficient background to easily follow the study. The Mat&Met and Results sections are detailed and concise. The Discussion and the conclusions are consistent with the molecular evidence collected in the study.
IMO, the choice to use sometimes the species’ common names, other times the scientific ones is not helpful and contribute to generate confusion in the readers. For this reason, I would recommend using the scientific names through the entire main text.
I found the methodological approach based on the combined use of SNPs and mtDNA brilliant and innovative. The PCA, the Neighbor Net and the Admixture analyses carried out on the nDNA have pointed out the presence of four main O. ammon clusters, one of them grouping individuals from two different subspecies (O. a. ammon and O. a. darwini). Such a result has evidenced once again the limit of the species classification based on traditional/morphological taxonomy and suggest the need to revise the classification of these two presumed subspecies. It is a crucial point to be further investigated in the future. Within the Central Asian populations sample, O. a. severtzovi has been found of admixed origin with a ratio O. a. polii / O. vignei = 75/25. The detection of a migration event from O. vignei to the common ancestor of O. a. polii / O. a. severtzovi (TreeMix analysis), combined with the results from mitogenomes analysis, has allowed the Author to infer that it was associated with migrations of urial males.
Although not decisive for the aims of the work, the molecular dating of the different mitochondrial lineages characterizing the radiation of the investigated O. ammon sub-species would have further improved the manuscript, which is nonetheless very valid.
I have just some minor changes to suggest in order to improve the manuscript. I have indicated them in the manuscript (pdf file)

Author Response
We thank our reviewer for valuable suggestions and helpful comments.
- Introduction, line 75: why only investigating 5 subspecies when 9 are defined? This would make more sense with the nowadays-accepted sub-speciation of argali (Introduction, line 71)
We investigated only the samples of Ovis ammon that we could collect so far. Unfortunately, we could not find samples of O. a. jubata, O. a. hodgsoni, O. a. nigrimontana and O. a. collium. I hope that this work will initiate collaborations with other researchers and help us investigate all the subspecies of argali.
- M&M, line 124: I am a bit surprised that data was collected by “trophy hunters”. Do you mean that, for example, the four 0. a. severtzovi - an endangered species - were killed for the purpose of this study? This sounds difficult to justify from an ethical point of view…
None of the animals were killed for the purpose of this study. None of the samples were obtained from poachers. We collaborated only with “trophy hunters” who had licenses that were given by national environmental organizations. Such organizations estimate the populations of wild animals and determine the number of licenses that could be issued. A limited number of licenses are being issued (not every year) even for endangered O. a. severtzovi. In fact trophy hunting can be beneficial for conservation of wild animals: “trophy hunters” are interested in old males which are not very important for a population but they pay a lot of money for licenses that could be used for fighting against poaching. And the greatest damage to populations of wild animals is caused by poachers who usually kill either young animals or females that are very important for conservation of populations
- M&M, line 136: what is the rationale for pruning SNP showing LD in this admixed sample? Stratification should lead to LD, which could be informative to distinguish distinct sub-species. For example, a SNP with a fixed allele in some subpopulations and the other allele fixed in the other subpopulations would be pruned (no heterozygous individuals) although it is informative for phylogenetic analyses.
LD pruning in processing of whole genome SNP data results in removing of loci that are in linkage disequilibrium i.e. inherited together. For example, if two loci are located close to each other on a chromosome and are inherited as a haplotype, one of these loci will be removed. Such LD pruning is even required by some programs. What you mention here is pruning of loci that are in Hardy-Weinberg equilibrium. And you are absolutely right that this filter should not be applied in the studies where different populations are investigated. In our work we did not use this filter.
- Results, line 224: I would have expected that the smallest population would have the greatest inbreeding coefficient, but it turns out that 0. severtzovi, which is seemingly the smallest in size, has the lowest inbreeding coefficient… How do you explain that? And could you not use ROH as a better estimate of the inbreeding coefficient? With small populations, inbreeding could be a concern, and precise estimation of F could help manage these populations.
We assume that the higher inbreeding coefficient in O. a. ammon was observed due to lower genetic diversity in this population. The census size of O. a. ammon is less than 5000 individuals which is not sufficiently larger than the census size of O. a. severtzovi – around 2000 individuals. We did not use ROH as in this study for investigation of O. ammon we used SNP-chip designed for domestic sheep – O. aries. Therefore, it can be used not for all the analyses, especially based on LD.
- Discussion, line 276: You show in you results and mention in the discussion that the diversity is lower in East Asian species than in Central Asian species. Do you have an explanation for this difference, or at least an hypothesis? It would be nice to have your thoughts on this in the paper, rather than simply mentioning that these results were similar to those obtained in Tserenbataa’s paper.
We added to Discussion: “We suggest that harsh climatic conditions and partial glaciation of Altai Mountains during the Last Glacial Maximum could lead to a decrease in genetic diversity in East Asian populations of argali.”
- Abstract, line 36: …that the 5 subspecies could be grouped into 4 clusters…
Done
- M&M, line 154: what do you mean by “we concatenated 2 rRNA and 13 protein coding genes”? Do you mean that you sequenced these 15 genes in addition to the mitochondrial DNA? If yes, this should not appear in the 2.3 paragraph, but rather in the 2.2 paragraph, and use something like: “In addition, we sequenced 2 rRNA and 13 protein coding genes to perform the phylogenetic analyses…”
It means that we did not use the entire mitogenome sequence for phylogenetic analysis, but only 2 rRNA and 13 protein coding genes. Therefore, we extracted them from the complete mitochondrial genome sequence and concatenated into one sequence. To make it more clear we made some changes in Manuscript: “To perform phylogenetic analysis we used sequences of concatenated 2 rRNA and 13 protein coding genes that were extracted from complete mitochondrial genomes and consisted of 13879 bp.”
- Results, line 177: “…to in the same cluster.”
Done
- Results, line 194: “…for the both distances...”
Done
- Results, Figure 3: please make the arrowheads more visible. Is there any explanation why the “yellow migration event” (migration of O. vignei to the common ancestor of O. polii and O. severtzovi) is not visible at all in O. polii’s genome (f.e. in Figure 2 C) ?
We made the arrowheads more visible. In Figure 3 we see a migration event from O. vignei to the common ancestor of O. polii and O. severtzovi with low migration weight. This most likely reflects that gene flow to this group was observed after their divergence from East Asian subspecies of argali. It was not detected by Admixture analysis since this event took place quite a long time ago, but pairwise genetic distances showed that Central Asian subspecies are genetically closer to O. vignei than East Asian subspecies.
- Discussion, line 307: maybe the existence of 0. a. humei should be mentioned in the introduction part. It is a bit strange that this possible distinction between subspecies mentions a potential subspecies that only appears in the discussion section.
We added to Introduction: “Two samples of O. a. karelini were collected in the area of Kyrgyz argali (O. a. humei) habitat, the subspecies that was described by Lydekkker in 1913, but is considered invalid by most researchers [Damm, 2014]. Since the habitat of this group of argali is located at the intersection of O. a. karelini and O. a. polii ranges, it was usually attributed to either the former [Valdez, 1982] or the latter [Wilson, 2005].”
Reviewer 2 Report
See in the attached file

Author Response
We thank our reviewer for valuable suggestions and helpful comments.
We accepted all the suggestions and recommendations.
- Replace with O. gmelini. Based on the last IUCN recommandation (see Michel & Ghoddousi, 2020) and following what suggested by Groves & Grubb (2011) and Hadjisterkotis & Lovari (2016), the correct name to designate the mouflon is Ovis gmelini
Ovis orientalis was replaced with Ovis gmelini
- In the main text the Authors refer to the different argali subspecies by using sometimes the scientific names, other times the common ones. Such a choice could generate confusion in the readers. I would suggest using the scientific names through the main text.
Common names were replaced with scientific names
- More recent Ref are available on this point. I would suggest to read the review entitled. "Sheep Post-Domestication Expansion in the Context of Mitochondrial and Y Chromosome Haplogroups and Haplotypes." Machová, K.; Málková, A.; Vostrý, L. Genes2022,13, 613. https://doi.org/10.3390/genes13040613
We substituted the link for divergence time between Ovis species.
- I would suggest using circles, squares, or other highlight colours to differentiate East Asian and Central Asian populations. It might help make reading easier.
The figure 5 was improved to make reading easier
- The Figure 1 could be further improved indicating also the distribution range of the argali and urial species. This would give more emphasis to the good job made by the authors in collecting samples.
Thank you for these suggestions, we will take them into account in our future work. At the moment we are not ready to create a map with exact distribution ranges. Before that, we would like to consult with specialists for clarification of distribution ranges of these species.